# Clinical Outcomes of Secondary Prophylactic Granulocyte Colony-Stimulating Factors in Breast Cancer Patients at a Risk of Neutropenia with Doxorubicin and Cyclophosphamide-Based Chemotherapy

**DOI:** 10.3390/ph14111200

**Published:** 2021-11-22

**Authors:** Jae Hee Choi, Min Jung Geum, Ji Eun Kang, Nam Gi Park, Yun Kyoung Oh, Sandy Jeong Rhie

**Affiliations:** 1Division of Life and Pharmaceutical Sciences Graduate School, Ewha Womans University, Seoul 03760, Korea; 20110700@kuh.ac.kr; 2Department of Pharmacy, Konkuk University Medical Center, Seoul 05030, Korea; coffee@kuh.ac.kr; 3Graduate School of Clinical Biohealth, Ewha Womans University, Seoul 03760, Korea; MJGEUM@yuhs.ac; 4Department of Pharmacy, Severance Hospital, Yonsei University Health System, Seoul 03722, Korea; 5College of Pharmacy, Ewha Womans University, Seoul 03760, Korea; jjadu@nmc.or.kr; 6Department of Pharmacy, National Medical Center, Seoul 04564, Korea; 7Graduate School of Pharmaceutical Sciences, Ewha Womans University, Seoul 03760, Korea; ngpark93@ewhain.net

**Keywords:** secondary prophylaxis, granulocyte colony-stimulating factor, breast cancer, neutropenia, adjuvant chemotherapy, doxorubicin, cyclophosphamide

## Abstract

Doxorubicin and cyclophosphamide (AC)-based chemotherapy has been a standard regimen for early-stage breast cancer (ESBC) with an intermediate risk (10–20%) of febrile neutropenia (FN). Secondary prophylaxis of granulocyte colony-stimulating factor (G-CSF) is considered in patients receiving AC-based chemotherapy; however, relevant studies are limited. Here, we retrospectively reviewed the electronic medical records of 320 patients who completed adjuvant AC-based chemotherapy from September 2016 to September 2020. Approximately 46.6% of the patients developed severe neutropenic events (SNE) during AC-based chemotherapy. Secondary prophylaxis of G-CSF reduced the risk of recurrent SNE (*p* < 0.01) and the relative dose intensity (RDI) < 85% (*p* = 0.03) in patients who had experienced SNE during AC-based chemotherapy. Age ≥ 65 years (*p* = 0.02) and alanine aminotransferase (ALT) or aspartate aminotransferase (AST) > 60 IU/L (*p* = 0.04) were significant risk factors for RDI < 85%. The incidences of FN, grade 4 neutropenia, unscheduled hospitalization, and interruption to the dosing regimen were reduced in patients administered secondary prophylaxis with G-CSF (before vs. after administration: FN, 19.4% vs. 4.6%; grade 4 neutropenia, 86.1% vs. 14.8%; unscheduled hospitalization, 75.9% vs. 11.1%; interruption to the dosing regimen, 18.5% vs. 8.3%). This study indicated the importance of active intervention of G-CSF use to prevent recurrent SNE and improve clinical outcomes in patients with breast cancer who receive AC-based chemotherapy.

## 1. Introduction

Adjuvant chemotherapy for patients with early-stage breast cancer (ESBC) reduces the risk of recurrence and mortality [1,2]. Doxorubicin-and-cyclophosphamide (AC)-based chemotherapy is the most accepted standard regimen for ESBC [3,4]. Febrile neutropenia (FN) and grade 4 neutropenia are serious hematologic adverse reactions to chemotherapy that contains myelotoxic agents. They increase the risk of rapid infection progress and may lead to death. The incidences of FN and grade 4 neutropenia in patients with AC-based chemotherapy range from 4.6–29.5% and 44.6–66.4%, respectively [5,6,7,8]. In many cases, prompt administration of prophylactic antibiotics and hospitalization are required until absolute neutrophil count (ANC) recovery [9,10].

The American Society of Clinical Oncology (ASCO), the European Organization for Research and Treatment of Cancer (EORTC), and the National Comprehensive Cancer Network (NCCN) guidelines recommend using primary prophylactic granulocyte colony-stimulating factor (G-CSF) when the risk of FN is >20% for all planned cycles of treatment [11,12,13]. The prophylactic use of G-CSF contributes to successful remission from breast cancer by maintaining ≥ 85% of the planned relative dose intensity (RDI) of chemo-medications during chemotherapy [14,15,16,17]. Moreover, the ASCO, EORTC, and NCCN guidelines define AC-based chemotherapy as an intermediate-risk (10–20%) regimen regarding FN and recommend considering secondary prophylaxis with G-CSF [11,12,13]. Previously, some researchers sought to evaluate the effect of secondary prophylaxis of G-CSF on reducing recurrent FN and maintaining RDI in patients with breast cancer who were treated with AC-based chemotherapy; however, the number of patient cases was limited, which reduced the statistical power [15,17,18]. Since September 2016, the National Health Insurance (NHI) in Korea has reimbursed the use of secondary prophylactic G-CSF for patients with breast cancer receiving AC-based chemotherapy. Therefore, it has become an affordable and common practice to administer prophylactic G-CSF in patients who experienced neutropenia. This is expected to improve patient outcomes. However, there are insufficient data to assess the impact of secondary prophylactic use of G-CSF in Korean patients with breast cancer.

We investigated the effects of secondary prophylactic administration of G-CSF and the risk factors of the recurrent FN, grade 4 neutropenia, and RDI < 85% in Korean patients with breast cancer who received AC-based chemotherapy.

## 2. Results

### 2.1. Patient Characteristics

There were 341 patients with breast cancer who completed adjuvant AC-based chemotherapy from September 2016 to September 2020. Twenty-one patients were excluded from the study because six were administered primary prophylactic G-CSF, nine had previous chemotherapy, four had hepatic or renal dysfunction, and two had missing values in laboratory tests. Thus, 320 patients were included and analyzed in this study. All patients were female, and the median age was 50 years. Most patients (86.9%) were in cancer stage I or II. One hundred and eight patients (33.8%) received the secondary prophylactic G-CSF of lipegfilgrastim (n = 63) or pegfilgrastim (n = 45). Another 65 patients (20.3%) received the G-CSF treatment of filgrastim, and the final 147 patients (45.9%) did not receive any G-CSF (Table 1).

### 2.2. The Overall Incidence of Severe Neutropenic Events and Risk Factors

Almost half of the patients (149/320, 46.6%) developed SNE due to AC-based chemotherapy (FN: 9.4%, n = 30; grade 4 neutropenia: 42.8%, n = 137; both: 5.6%, n = 18). However, 12 patients were excluded because they developed FN or grade 4 neutropenia during the last chemotherapy cycle. Thus, 108 of the 137 patients were confirmed to have received secondary prophylactic G-CSF due to AC-based chemotherapy.

Multivariable analysis revealed that cancer stage III and diabetes mellitus (DM) were significant risk factors of FN (cancer stage III: odds ratio (OR) 4.20, 95% confidence interval (CI) 1.75–10.06; DM: OR 3.61, 95% CI 1.24–10.44). In grade 4 neutropenia, cancer stage III and basal Hb < 12 g/dL were significant risk factors (cancer stage III: OR 8.90, 95% CI 3.74–21.07; basal Hb < 12 g/dL: OR 2.23, 95% CI 1.14–4.38). Moreover, when we performed multivariable analysis to evaluate the risk factors of SNE in total, cancer stage III and basal Hb < 12 g/dL were significant risk factors of SNE (cancer stage III: OR 13.96, 95% CI 4.82–40.46; basal Hb < 12 g/dL: OR 2.41, 95% CI 1.21–4.78) (Table 2).

### 2.3. Recurrence of Severe Neutropenic Events and Its Risk Factors

The recurrence of SNE was 27.0% (n = 37/137) (FN: 5.1%, n = 7; grade 4 neutropenia: 24.1%, n = 33, both: 2.2%, n = 3) (Figure 1). Secondary prophylaxis of G-CSF was associated with reducing the risk of recurrent SNE (OR 0.17, 95% CI 0.07–0.43) (Table 3).

### 2.4. RDI after Experiencing Severe Neutropenic Events

The mean RDI for patients who had SNE during overall chemotherapy was 94.1% in our study. RDI < 85% was present in 12.4% (17/137) of these patients. In multivariable analysis, age ≥ 65 years, ALT or AST > 60 IU/L elevation were the risk factors of RDI < 85% (age ≥ 65 years: OR 11.78, 95% CI 1.51–91.64; ALT or AST > 60 IU/L elevation: OR 3.51, 95% CI 1.09–11.37). Secondary prophylaxis of G-CSF reduced the risk of RDI < 85% (OR 0.27, 95% CI 0.08–0.86) (Table 4).

### 2.5. Clinical Outcomes of Secondary Prophylaxis of G-CSF

The incidence of FN was reduced significantly after the administration of G-CSF (before G-CSF: 19.4%, n = 21; after G-CSF: 4.6%, n = 5, *p* < 0.01). Moreover, the incidence of grade 4 neutropenia significantly decreased after the administration of G-CSF (before G-CSF: 86.1%, n = 93; after G-CSF: 14.8%, n = 16, *p* < 0.01).

Unscheduled hospitalization due to SNE in the patients receiving secondary prophylactic G-CSF was 75.9% (n = 82) before administration. This decreased significantly to 11.1% (n = 12) after administration (*p* < 0.01). Interruption to the dosing regimen due to SNE also decreased significantly, from 18.5% (n = 20) before administration to 8.3% (n = 9) after administration (*p* < 0.01) (Figure 2).

## 3. Discussion

In this study, we confirmed that the use of secondary prophylaxis of second-generation G-CSF was associated positively with clinical outcomes. We found a decrease in recurrent SNE, unscheduled hospitalization, and interruptions to the chemotherapy regimen in Korean patients with ESBC who received adjuvant AC-based chemotherapy. Over the past decades, adjuvant anthracycline-based chemotherapy has been effective and safe in treating patients with breast cancer [3,4]. AC-based regimens (such as doxorubicin, 60 mg/m^2^, and cyclophosphamide, 600 mg/m^2^ every 3 weeks), alone or followed by docetaxel, are associated with great risk reduction, such as low myelosuppression. Therefore, they remain the choice adjuvant chemotherapy in patients with EBSC who have any level of risk [3,4,19,20]. Guidelines by the ASCO, EORTC, and NCCN recommend that a secondary prophylaxis of G-CSF should be considered in patients who have experienced FN and dose-limiting neutropenic events in intermediate-risk-group chemotherapy [11,12,13]. However, studies that have focused on its effect in patients with AC-based chemotherapy in clinical practice are limited. Previous studies have analyzed the effects of secondary prophylaxis of G-CSF on preventing dose reduction in patients with breast cancer treated using different adjuvant chemotherapies [15,17]. They have shown that this treatment regimen can delay the time until recurrent neutropenic events in different solid cancers, such as breast, colorectal, lung, and ovarian cancers [18].

Hepatic and renal dysfunction are risk factors of FN [5,21,22]. Only four patients in this study had hepatic and renal impairments at baseline. We excluded these from our analysis because they were not eligible for the full dose of chemotherapy, or because their chemo-schedule was postponed based on the protocol in our institution.

Cancer stage III, DM, and basal Hb < 12 g/dL were risk factors for FN and grade 4 neutropenia in this study. In line with this, advanced cancer stage has been reported as a risk factor for FN in two prediction model studies which included patients with breast cancer [23,24], as well as in a case series study with patients with non-Hodgkin’s lymphoma who received initial CHOP chemotherapy [25]. Supporting evidence to explain this association remains weak. The prediction model study suspected that advanced cancer stage and disease severity might reflect the rapid and early initiation of chemotherapy [23]. Unfortunately, most of our population started chemotherapy approximately four weeks after surgery; thus, this may not apply to our study. DM is associated with an increased risk of chemotherapy-induced neutropenia [26]. Uncontrolled DM, which may lead to unhealed wounds and poorly resolved infections, can increase one’s risk of FN and its complications [27]. Furthermore, we found a higher rate of unscheduled hospitalization due to SNE in patients with DM than in those without DM.

Basal Hb < 12 g/dL was a significant risk factor for grade 4 neutropenia in our study. A previous study has reported that pre-existing anemia may increase the risk of developing FN or severe neutropenia in patients treated with chemotherapy [28]. Lower levels of hemoglobin are associated with myelosuppression, which can further exacerbate the condition and lead to a higher incidence of FN [29]. This may explain why basal Hb < 12 g/dL increases the incidence of grade 4 neutropenia.

The administration rate of secondary prophylaxis of G-CSF was 78.8% (108/137) in our study, which was higher than the 66.7% (14/21) of Japanese patients with breast cancer who received an epirubicin and cyclophosphamide (EC) regimen from 2014 to 2018 [14]. This can be explained by the introduction of reimbursement for secondary prophylaxis G-CSF treatment by the Korea National Health Insurance (NHI) since September 2016. This treatment is not reimbursed in Japan. The recurrent SNE in the non-administered group of secondary prophylactic G-CSF was higher than in the administered group. This suggests that the secondary prophylaxis of G-CSF can protect against recurrent SNE. Additionally, after secondary prophylaxis of G-CSF, interruptions in the dosing regimen caused by SNE were significantly reduced. Secondary prophylaxis of G-CSF was effective in maintaining an adequate RDI, which reduced the recurrence of neutropenic events. This is consistent with the results of previous studies [15,17]. In this study, the mean RDI of patients with SNE was 94.1%; RDI < 85% was present in 12.4% (17/137) of these patients. The risk factors for RDI < 85% were age ≥ 65 years and elevated ALT or AST > 60 IU/L.

The risk of chemo-related adverse drug reactions increases as major organ function deteriorates in the elderly [30]; therefore, old age is a risk factor for RDI < 85% [4,31]. In our study, patients aged 65 years or older received a 10–25% dosing reduction during chemotherapy and were hospitalized for an average of 9.5 days due to chemotherapy-related adverse reactions, which was longer than the average of 4.0 days for younger patients. Doxorubicin and cyclophosphamide are metabolized in the liver; therefore, dose adjustments are required in hepatic dysfunction [32,33]. These properties of anticancer drugs have been shown in prospective studies that include patients with breast cancer; therefore, grade 2 and higher ALT or AST levels are defined as hepatic dysfunction [18,34]. In these cases, chemotherapy cycles are suspended until blood chemistry abnormalities are resolved to normal or grade 1. The most frequent cause of chemotherapy delay is transaminase elevation [34]. To the best of our knowledge, an association between transaminase elevation and risk of RDI < 85% has seldom been reported. It is reasonable to suggest close monitoring of sudden changes and upward trending of transaminase, especially for ALT or AST > 60 IU/L, prior to chemotherapy. This would promote the administration of a full dose of chemotherapy without any reductions or delays.

This is a retrospective study using medical records from a single institution. We found that cancer stage III was a significant risk factor for FN; however, it was difficult to identify a pathological mechanism because our study only investigated medical records. Thus, further ex vivo studies will be required to elucidate the underlying mechanisms. We analyzed the risk factors for recurrence of SNE, but not FN and grade 4 neutropenia. Our study was conducted by collecting the medical records of 320 patients, which was not a small scale compared with previous studies; however, it was difficult to analyze the risk factors for each SNE. Therefore, larger studies will be required to analyze the risk factors of both recurrent FN and grade 4 neutropenia.

Despite these limitations, this study confirmed the most recent clinical practice trends from 2016 to 2020. Moreover, a similar study including patients who were being treated with different chemotherapies for several solid cancers reported a positive effect of secondary prophylaxis of G-CSF on decreased neutropenic events [18]. This is too generalized to apply these results to patients who received AC-based chemotherapy for breast cancer. Nonetheless, we conducted this study to evaluate the clinical outcomes of secondary prophylaxis of G-CSF in a sufficient number of patients with breast cancer who were treated with AC-based chemotherapy. Taken together, we found that secondary prophylaxis of G-CSF improved clinical outcomes; patients showed decreased rates of recurrent SNE, unexpected hospitalizations, and interruptions to the chemo-regimen.

## 4. Materials and Methods

### 4.1. Patients

We retrospectively reviewed the electronic medical records (EMR) of patients with breast cancer who were ≥18 years and had completed a course of AC-based chemotherapy as the adjuvant therapy from September 2016 to September 2020 at theKonkuk University Medical Center. Patients were excluded if they received primary prophylactic G-CSF, started AC-based chemotherapy at other hospitals, had received previous chemotherapy, had hepatic or renal dysfunction at baseline, or had missing values in laboratory tests.

Adjuvant AC-based chemotherapy in the routine clinical practice included AC and AC-D regimens. In general, patients on the AC regimen received doxorubicin at 60 mg/m^2^ intravenously on day 1 and cyclophosphamide at 600 mg/m^2^ intravenously on day 1 over 3 weeks for 4 cycles. On the AC-D regimen, patients received doxorubicin at 60 mg/m^2^ intravenously on day 1 and cyclophosphamide at 600 mg/m^2^ intravenously on day 1 over 3 weeks for 4 cycles followed by docetaxel at 75 mg/m^2^ on day 1 over 3 weeks for 4 cycles. Secondary prophylaxis of G-CSF was defined as the administration of G-CSF to a patient who had experienced FN or grade 4 neutropenia to reduce their risk of neutropenia from subsequent chemotherapy. For secondary prophylaxis, lipegfilgrastim or pegfilgrastim was administered subcutaneously at a fixed dose of 6 mg once at 24 h after completion of each chemotherapy cycle

### 4.2. Study Design

We collected the following patient information: age, sex, height, weight, body mass index (BMI), body surface area (BSA), estrogen receptor (ER) status, progesterone receptor (PR) status, human epidermal growth factor receptor 2 (HER2) status, cancer stage, comorbidities (cardiovascular diseases, diabetes mellitus, thyroid diseases, and rheumatoid diseases), menopause status, use of G-CSF, type of G-CSF, and laboratory values.

The incidence and risk factors of SNE after AC-based chemotherapy were analyzed. SNE referred to both FN and grade 4 neutropenia in this study. FN was defined as when patients had a single temperature > 38.3 °C or ≥38 °C for over 1 h and an ANC < 1000 cell/mm^3^. Grade 4 neutropenia was defined as an ANC < 500 cells/mm^3^, as previously reported in the Common Terminology Criteria for Adverse Events (CTCAE) version 5.0, or via diagnosis at the EMR by a physician in this study.

Additionally, the incidence and risk factors of recurrent SNE were assessed. RDI after experiencing SNE and the risk factors for low RDI (<85%) were assessed. Clinical outcomes were the occurrence of FN, grade 4 neutropenia, unscheduled hospitalization, and the interruptions to the dosing regimen (by dose reduction or dosing delay) of chemotherapy due to SNE.

### 4.3. Statistical Analysis

Categorical data were compared using the chi-squared or Fisher’s exact tests. McNemar’s test was used to compare the incidence of FN, grade 4 neutropenia, unscheduled hospitalization, and interruption to the dosing regimen of chemotherapy before and after administration of secondary prophylactic G-CSF. Logistic regression analysis was used to identify risk factors for FN, grade 4 neutropenia, and relative dose intensity (RDI) < 85%. RDI (%) was calculated with the equation: [actual total dose per week (mg/m^2^/week)]/[standard planned total dose per week (mg/m^2^/week)] [14]. Variables with *p* < 0.20 in the univariable analysis were entered into the multivariable logistic analysis. All statistical analyses were conducted using IBM SPSS^®^ Statistics 28.0, and *p* < 0.05 was considered statistically significant.

## 5. Conclusions

Approximately 46.6% of patients experienced SNE during AC-based chemotherapy. Secondary prophylaxis of G-CSF reduced the recurrence of SNE and maintained RDI ≥ 85%. Further, it reduced the number of unscheduled hospitalizations and interruptions to the dosing regimen in patients on AC-based chemotherapy for breast cancer.

## Figures and Tables

**Figure 1 pharmaceuticals-14-01200-f001:**
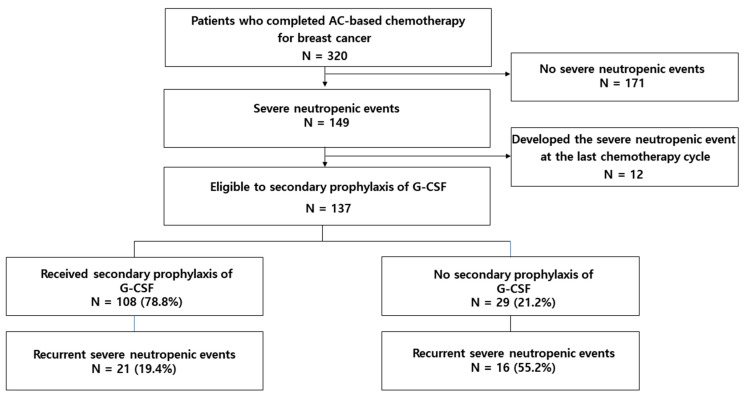
Secondary prophylaxis of G-CSF and recurrent severe neutropenic events. AC, doxorubicin and cyclophosphamide; G-CSF, granulocyte colony-stimulating factor.

**Figure 2 pharmaceuticals-14-01200-f002:**
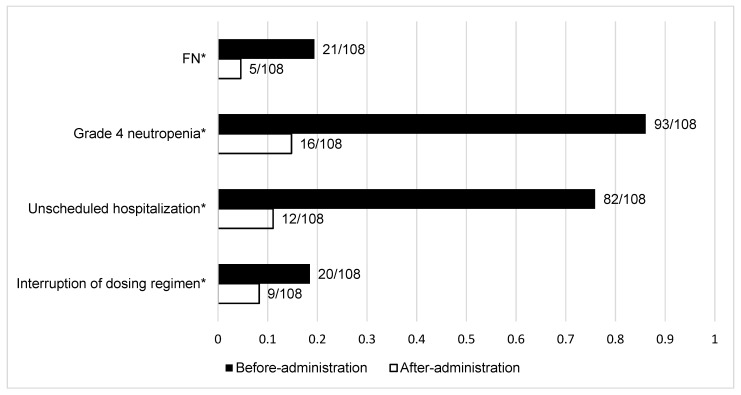
Clinical outcomes of secondary prophylaxis of G-CSF. FN: febrile neutropenia; G-CSF: granulocyte colony-stimulating factor. * *p* < 0.01.

**Table 1 pharmaceuticals-14-01200-t001:** Patient characteristics (n, %) (N = 320).

Characteristics	Number (%)
Age (years), median (IQR)	50 (45–58)
Body mass index (kg/m^2^), median (IQR)	23.8 (21.7–26.2)
Body surface area (m^2^), median (IQR)	1.6 (1.5–1.7)
ER status	
Negative	58 (18.1%)
Positive	262 (81.9%)
PR status	
Negative	149 (46.6%)
Positive	171 (53.4%)
HER2 status	
Negative	275 (85.9%)
Positive	45 (14.1%)
Cancer stage	
I	97 (30.3%)
II	181 (56.6%)
III	42 (13.1%)
Comorbidities	
Cardiovascular diseases	54 (16.9%)
Diabetes mellitus	25 (7.8%)
Thyroid diseases	20 (6.3%)
Rheumatoid diseases	3 (0.9%)
Menopause status	
Pre-menopause	177 (55.3%)
Post-menopause	143 (44.7%)
Use of G-CSF	
Secondary prophylaxis	108 (33.8%)
Lipegfilgrastim	63 (58.3%)
Pegfilgrastim	45 (41.7%)
Treatment	65 (20.3%)
None	147 (45.9%)
Basal laboratory values, median (IQR)	
WBC count (×10^3^/μL)	6.2 (5.2–7.2)
ANC (cells/mm^3^)	3270 (2530–4130)
Hemoglobin (g/dL)	13.1 (12.3–13.7)
Platelet count (×10^3^/μL)	249.0 (218.3–295.0)
AST (IU/L)	24.0 (20.0–28.0)
ALT (IU/L)	17.0 (13.0–24.8)

ANC, absolute neutrophil count; ALT, alanine aminotransferase; AST, aspartate aminotransferase; ER, estrogen receptor; G-CSF, granulocyte colony-stimulating factor; HER2, human epidermal growth factor receptor 2; IQR, interquartile range; PR, progesterone receptor; WBC, white blood cell.

**Table 2 pharmaceuticals-14-01200-t002:** Risk factors for febrile neutropenia, grade 4 neutropenia, and severe neutropenic events.

Category	Univariable Analysis		Multivariable Analysis	
Variables	Odds Ratio (95% CI)	*p**-*Value	Odds Ratio (95% CI)	*p**-*Value
FN (N = 320)				
BSA ≥ 1.65 (vs. <1.65)	0.55 (0.23–1.33)	0.19	0.45(0.18–1.14)	0.09
Cancer stage: III over I–II	4.03 (1.73–9.37)	<0.01	4.20(1.75–10.06)	<0.01
DM present over absent	3.57 (1.30–9.77)	0.01	3.61(1.24–10.44)	0.02
Grade 4 Neutropenia (N = 320)				
ER: positive over negative	1.53 (0.85–2.78)	0.16	1.78(0.92–3.44)	0.09
Cancer stage: III over I–II	8.63 (3.70–20.13)	<0.01	8.90(3.74–21.07)	<0.01
Basal Hb < 12 g/dL	2.46 (1.30–4.65)	0.01	2.23(1.14–4.38)	0.02
Basal WBC count < 4000/μL	3.72 (0.97–14.30)	0.06	2.98(0.71–12.61)	0.14
Severe Neutropenic Events (N = 320)				
Cancer stage: III over I–II	14.29(4.96–41.17)	<0.01	13.96(4.82–40.46)	<0.01
Basal Hb < 12 g/dL	2.55(1.33–4.87)	<0.01	2.41(1.21–4.78)	0.01
Basal WBC count < 4000/μL	3.18 (0.83–12.20)	0.09	2.51(0.59–10.70)	0.21

BSA, body surface area; DM, diabetes mellitus; ER, estrogen receptor; FN: febrile neutropenia, Hb, hemoglobin; WBC, white blood cell.

**Table 3 pharmaceuticals-14-01200-t003:** Risk factors for recurrent severe neutropenic events.

Variables	Univariable Analysis		Multivariable Analysis	
	Odds Ratio (95% CI)	*p-*Value	Odds Ratio (95% CI)	*p-*Value
Age ≥ 65 years (vs. <65)	0.67 (0.07–6.17)	0.72	-	
BMI ≥ 25 (vs. <25)	1.61 (0.75–3.46)	0.22	-	
BSA ≥ 1.65 (vs. <1.65)	0.85 (0.38–1.90)	0.70	-	
ER: positive over negative	0.82 (0.31–2.18)	0.69	-	
PR: positive over negative	1.40 (0.65–3.03)	0.39	-	
HER2: positive over negative	1.22 (0.46–3.27)	0.69	-	
Cancer stage: III over I–II	1.20 (0.52–2.77)	0.66	-	
CV diseases present over absent	2.10 (0.84–5.21)	0.11	1.58 (0.58–4.29)	0.37
DM present over absent	0.79 (0.21–3.06)	0.74	-	
Post-menopause over pre-menopause	1.69 (0.79–3.62)	0.17	2.02 (0.85–4.80)	0.11
Basal Hb < 12 g/dL	0.78 (0.30–2.01)	0.61	-	
Basal WBC count < 4000/μL	0.44 (0.05–3.74)	0.45	-	
Basal ALT or AST > 40 IU/L	0.69 (0.21–2.22)	0.53	-	
Secondary prophylaxis of G-CSF	0.20 (0.08–0.47)	<0.01	0.17 (0.07–0.43)	<0.01

ALT, alanine aminotransferase; AST, aspartate aminotransferase; BMI, body mass index; BSA, body surface area; CV, cardiovascular; DM, diabetes mellitus; ER, estrogen receptor; G-CSF, granulocyte colony-stimulating factor; Hb, hemoglobin; HER2, human epidermal growth factor receptor 2; PR, progesterone receptor; WBC, white blood cell.

**Table 4 pharmaceuticals-14-01200-t004:** Risk factors for RDI < 85% of patients with severe neutropenic events.

Variables	Univariable Analysis		Multivariable Analysis	
	Odds Ratio (95% CI)	*p-*Value	Odds Ratio (95% CI)	*p-*Value
Age ≥ 65 years (vs. <65)	5.20 (0.80–33.68)	0.08	11.78 (1.51–91.64)	0.02
BMI ≥ 25 (vs. <25)	1.38 (0.50–3.83)	0.54	-	
BSA ≥ 1.65 (vs. <1.65)	1.01 (0.35–2.93)	0.98	-	
ER: positive over negative	3.59 (0.45–28.54)	0.23	-	
PR: positive over negative	1.13 (0.40–3.17)	0.82	-	
Cancer stage: III over I–II	1.15 (0.37–3.51)	0.81	-	
CV diseases present over absent	2.08 (0.66–6.57)	0.21	-	
DM present over absent	0.56 (0.07–4.62)	0.59	-	
Thyroid diseases present over absent	0.88 (0.10–7.47)	0.90	-	
Post-menopause over pre-menopause	1.47 (0.53–4.07)	0.46	-	
Basal Hb < 12 g/dL	0.74 (0.20–2.76)	0.65	-	
Basal WBC count < 4000/μL	1.19 (0.13–10.51)	0.88	-	
Basal ALT or AST > 40 IU/L	3.15 (0.97–10.30)	0.06	2.99 (0.81–11.07)	0.10
ALT or AST > 60 IU/L elevation	3.06 (1.08–8.70)	0.04	3.51 (1.09–11.37)	0.04
Recurrence of SNE	0.54 (0.15–2.01)	0.36	-	
Secondary prophylaxis of G-CSF	0.32 (0.11–0.94)	0.04	0.27 (0.08–0.86)	0.03

ALT, alanine aminotransferase; AST, aspartate aminotransferase; BMI, body mass index; BSA, body surface area; CV, cardiovascular; DM, diabetes mellitus; ER, estrogen receptor; G-CSF, granulocyte colony-stimulating factor; Hb, hemoglobin; PR, progesterone receptor; SNE: severe neutropenic events; WBC, white blood cell.

## Data Availability

The data presented in this study are available on request from the corresponding author. The data are not publicly available due to privacy restrictions and patient confidentiality. This study used and rearranged part of the data for J.H.C.’s 2021 Ph.D. thesis.

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
