# Peer review of "Clinical Outcomes of Secondary Prophylactic Granulocyte Colony-Stimulating Factors in Breast Cancer Patients at a Risk of Neutropenia with Doxorubicin and Cyclophosphamide-Based Chemotherapy"

_pharmaceuticals, 2021, doi:10.3390/ph14111200_

Round 1

Reviewer 1 Report

Authors present a manuscript addressing the clinical outcomes of secondary prophylactic granulocyte colony-stimulating factors in breast cancer patients at a risk of neutropenia with doxorubicin and cyclophosphamide-based chemotherapy. The aim of the study was to investigate the effects of secondary prophylactic administration of G-CSF and the risk factors of the recurrent FN, grade 4 neutropenia, and RDI <85% in Korean patients with breast cancer who received AC-based chemotherapy. The topic of the article is relevant in the field. I my opinion the manuscript is very well-written. Obviously, I endorse to publish this article once Authors correct my suggestions:

Minor points:

1) Please improve the font size in figure 1.

2) Minor modification of the grammar is required.

Reviewer 2 Report

The authors tried to identify if G-CSF administration can prevent neutropenia events, but there are some questions regarding the analysis.

  1. The age of the patients was between 45-58, but in table 3 and 4  they analyze data from patients over 65 years compared to under 65 years old? How is this possible if you did not have any patients with this age?
  2. Why there were analyzed only EP patients and not other patients? There are several studies that demonstrated that TNBC patients have higher risk of neutropenia.
  3. The number of patients with stage III breast cancer is to small compared to the other stages, more patients should be included with this stage.
  4. Also, have you consider analyzing patients with obesity to see how they respond to this?

Round 2

Reviewer 2 Report

-